# Antibacterial property of lead telluride quantum dot layer fabricated on glass substrate

**Samuel Onuh Abuh**[1]◉, **Svetlana Lyssenko**[1]◉, **Ayan Barbora**[1], **Iryna Hovor**[2], **Faina Nakonechny**[2], **Refael Minnes**[1]*

1 Department of Physics, Faculty of Natural Sciences, Ariel University, Ariel, Israel, 2 Department of Chemical Engineering, Faculty of Natural Sciences, Ariel University, Ariel Israel

◉ These authors contributed equally to this work.
* refaelm@ariel.ac.il

## Abstract

Lead Telluride (PbTe) is a narrow band gap semiconductor alloy with excellent thermoelectric properties for several energy harvesting applications. However, the antibacterial properties of PbTe quantum dots (QDs) have not been investigated. PbTe QDs were synthesized using simple spin-coating method and deposited on Titanium dioxide layered ITO glass substrates. The resulting layers of PbTe QDs on the substrates were characterized using high-resolution scanning electron microscope, energy-dispersive X-ray spectroscopy, Fourier transform infra-red spectroscopy and contact angle measurement. The characterization results showed thin layers of PbTe quantum dots with mean sizes 6.1 ± 0.5 nm, 9.8 ± 0.7 nm, and 13.2 ± 1.1 nm and reduced surface wettability. PbTe QDs were tested for their antibacterial activity against Gram-positive bacteria *Staphylococcus aureus* and Gram-negative *Escherichia coli*, *Salmonella Paratyphi B* and *Pseudomonas aeruginosa.* The antibacterial effect of the QDs was estimated using the zones of inhibition to bacterial growth. The results show excellent antibacterial activity of PbTe QDs towards Gram-negative bacteria. FTIR micro-spectroscopy suggests disruption of cell boundaries as possible mechanism of antibacterial action of PbTe QDs. Given the demonstrated antibacterial effectiveness, the PbTe QDs can be considered for nanocoating bacterial-prone surfaces like solar panels to minimize bacterial colonization and improve system performance.

## 1. Introduction

Bacterial proliferation on surfaces, culminating in the development of colonies and subsequent biofilm formation, remains an ongoing and persistent problem in current times. Microbial colonization continues to pose great risk in human healthcare [1] as well as in industries including food, marine and water systems, and in domestic settings [2], causing infection, fouling, corrosion, and a reduction in the cost efficiency of

**Data availability statement:** All relevant data are within the manuscript and its Supporting Information files.

**Funding:** The author(s) received no specific funding for this work.

**Competing interests:** The authors have declared that no competing interests exist.

systems. Recent advancements in targeted therapies, such as non-ionizing radiation techniques for antiviral treatments during viral pandemics [3] highlight the critical role of innovative approaches in addressing infectious diseases. Similarly, exploring novel materials like quantum dots for antibacterial applications offers new strategies in managing bacterial colonization [4]. However, their application for antibacterial purposes is limited by toxicity and environmental concerns [5–8]. It has been proposed that size modification, encapsulation, and surface functionalization of metallic NPs [9–11] can minimize their toxic effect while allowing for their beneficial use. Various studies have examined the antibacterial activity of NPs and their composites, including $Cu_2SnS_3$, Ag, Au, ZnO, MgO, $BaTiO_3$ and so on [12–17]. Quantum dots (QDs) including carbon, graphene, among others have also been tested for their antibacterial properties [18]. However, many of these studies demonstrated antibacterial activities through NPs deposited on filter papers, direct use of sol-gel solutions of NPs against bacteria or using NPs suspended in solvents at varying concentrations. Even though these methods showed notable bactericidal effects, methods involving surface engineering for antibacterial purposes are of important practical interest since bacteria in their sessile state constitute a significant contribution to biofouling of environmental surfaces such as those of underwater pipelines, solar panels as well as medical materials like catheters. Hence, researchers have employed different designs of antibacterial surfaces including bactericidal, anti-adhesion, and mechano-bactericidal approaches, utilizing nanomaterials such as copper oxide, silver, and zinc as coatings for such surfaces in order to minimize bacterial colonization and biofilm formation [19].

Although considerable progress has been made in the fabrication of bioactive surfaces, some limitations exist including coating stability and its long-term durability in terms of resistance to weathering and abrasive stress [20,21]. It therefore remains an ongoing effort to design bioactive surfaces that can offer application specific protection against biofouling, while maintaining desirable levels of chemical and mechanical stability, such as resistance to wear and corrosion [22], making multi-property materials candidates of great interest in antimicrobial research.

Lead Telluride (PbTe) is a narrow band gap semiconductor alloy that has been shown to possess excellent thermoelectric properties for various applications, including energy harvesting power generation, thermal sensing equipment, and solar cells [23–27]. Despite these applications, the antibacterial properties of PbTe QDs remain unexplored. While lead is a toxic metal, it has anti-corrosion property. Knowing the antibacterial potential of its alloy, PbTe, at the quantum dot scale may therefore be crucial for multi-purpose, tailored, and application-specific uses, especially in certain industrial settings where human exposure to the active coating materials is limited and/or controlled. Here, PbTe QD layers were prepared using the rapid, low-cost, and simple spin-coating method in accordance with the solid-state ligand exchange (LE) for layer stabilization to test their antibacterial effect. The multifunctionality of nanomaterials is well demonstrated by recent advancements in treating human metastatic melanoma with molecular chimeras like CM358 [28]. Similarly, PbTe quantum dots exhibit both antibacterial and potential electronic properties [29], offering a dual advantage in industrial applications, for example, in nano-coating photovoltaic panels or the interior of power plant tubes [30].

Photovoltaic panels have been reported as susceptible to bacterial colonization, resulting in a decrease in their solar conversion efficiency [31,32]. The aim of this work is to evaluate the antibacterial activity of PbTe QD layers fabricated on ITO glass substrates and functionalized with a short organic linker for enhanced stability, against representatives of Gram-positive and Gram-negative bacteria. Our investigation serves to provide baseline data on the antibacterial properties of PbTe quantum dots, which could pave way for additional, tailored applications for the semiconductor material.

## 2. Materials and methods

### 2.1. Materials

Tri-n-octyl phosphine (TOP) (90% tech.), Te powder (99.999%), squalane (99%), lead acetate trihydrate ($Pb(C_2H_3O_2)_{2*3}(H_2O)$), (99%), oleic acid (OA) (90% tech), and p-phenylene diamine (PDA) (97 + %) were obtained from Holand Moran; methanol (AR), and hexane (95%), were purchased from BioLab; ethanol (99.5%) and isopropanol (<99.8% tech.) were purchased from Romical; indium-titanium oxide (ITO) glass substrates, helmanex III were obtained from Osilla; Ti-Nanoxide BL/SC (Solaronix); deionized water (> 18.0 MΩcm$^{-1}$). All chemicals were used as purchased without further purification.

### 2.2. Synthesis of PbTe QDs

Details of PbTe QDs synthesis and layer fabrication method as used in our experiments have been reported previously [33,34]. 6.1±0.5 nm PbTe QDs were synthesized in a three-neck round-bottom flask under $N_2$ gas using the standard Schlenk line technique. 570 mg (1.5 mmol) of Pb acetate trihydrate ($Pb(C_2H_3O_2)_{2*3}(H_2O)$), 1 mL (3.2 mmol) of oleic acid (OA), 14 mL of squalene, and 6 mL of 0.5 M tellurium-tri-n-octyl phosphine (TOP-Te) solution were used. The TOP-Te solution was stirred overnight until all the tellurium powder was completely dissolved, and the solution turned transparent and yellow. Pb salt, OA, and squalene were mixed and heated to 40°C under vacuum for 15 minutes. Subsequently, the reaction mixture was heated to 100°C and kept at this temperature for an additional 45 minutes. The vacuum was replaced with $N_2$ gas, and the mixture was heated to 180°C. 6 mL of a 0.5 M TOP-Te solution was rapidly injected into the reaction mixture. The reaction temperature dropped to ~155°C and was held at this temperature (± 2°C) for 2 minutes. The reaction was quenched in a water bath to room temperature. The mixture was washed with 10 mL of hexane in a 50 mL centrifuge tube. The particles were centrifuged for 5 min at 4000 rpm. Further, 20 mL of ethanol was added, and the tube was centrifuged for 5 min at 4000 rpm. The supernatant was disposed of, and the particles were washed with ethanol and redispersed in 10 mL of hexane. The 9.8±0.7 nm and 13.2±1.1 nm QDs were synthesized with the same protocol, changing the OA volume and TOP-Te concentration. For 9.8 nm QD synthesis, 3 mL of OA, 12 mL of squalene and 6 mL of 0.75 M TOP-Te were added to the reaction. For 13.2 nm QD synthesis, 6 mL of OA, 9 mL of squalene, and 6 mL of 0.75 M TOP-Te were used.

### 2.3. Layers fabrication solid-state ligand exchange

The ITO glass substrates were first cleaned using a supplier-recommended technique.. Ti-nanoxide ($TiO_2$) deposition on the ITO glass substrate was performed in a $N_2$ glove box. $TiO_2$ lowers the surface roughness on ITO and promotes fabrication of uniform monolayers of thin film [33]. 50 $\mu$l of Ti-nanoxide solution was drop-casted, in a static regime, on the ITO glass substrates and spun in a spin coater at 5000 rpm for 30 seconds. The deposited $TiO_2$ layers were annealed at 550°C for one hour under 6 L/min air flow in a tube furnace with a heat acceleration of 5°C/min. This step marked the end of the fabrication process for the $TiO_2$ on ITO ($TiO_2$/ITO) glass substrates used as control. Further, 80 $\mu$l PbTe QDs with a concentration of 20 mg/mL were drop-casted on a $TiO_2$/ITO glass substrate in the static regime and spun at 3000 rpm spinning speed for 30 seconds. The prepared PbTe on $TiO_2$/ITO (PbTe/$TiO_2$/ITO) layers were dried under air for 30 minutes for further use. At the end of this stage, some samples were set aside (and marked 'No LE') to test the effect of the QDs without the organic ligand.

Solid-state ligand exchange (LE) was performed in 0.1 M PDA in MeOH solution. The fabricated PbTe/ TiO$_2$/ITO layer was placed in a Petri dish containing 5 mL of 0.1 M PDA, ensuring that the PDA solution fully enveloped the entire substrate. Solid-state LE was performed for 5 minutes. After rinsing the fabricated layer with methanol three times to remove the excess PDA, the layer was dried using N$_2$ flow. PbTe QD deposition and solid-state LE were performed two times on each TiO$_2$/ITO glass substrate to ensure full substrate surface coverage. A schematic diagram of the synthesis of PbTe quantum dots and the layer fabrication is presented in Fig 1.

## 2.4. Layers characterization

All samples were analysed using a high-resolution scanning electron microscope (HR-SEM) (Ultra-High-Resolution Maia 3 FE-SEM, Tescan). The QDs size and morphology were analysed using a STEM (bright) detector and an electron beam voltage of 25–30 kV. Samples were prepared by drop casting on a copper grid. The particle size and count generator program was used for PbTe-QDs size calculations [35]. All the image processing was performed by the threshold function. Energy-dispersive X-ray spectroscopy (EDS) analysis of the layers was obtained by the Aztec microanalytic system (Oxford Instruments). UV-Vis transmission and FTIR spectroscopic studies were carried out using Jasco V-730 and FTIR 6800 FV spectrometers, respectively. Contact angle measurements were carried out using a digital microscope (HD multi-function digital microscope). The contact angle of 2 μL of double-distilled water on the PbTe layer surfaces was measured at four different locations and averaged to give an estimate of the surface wettability.

## 2.5. Antibacterial assay

Cultures of Gram-positive bacteria *Staphylococcus aureus* and Gram-negative bacteria *Escherichia coli*, *Salmonella Paratyphi B* and *Pseudomonas aeruginosa* were grown on Brain Heart agar plates (BHA, Acumedia, Lansing, MI, USA) for 24 h, transferred into Brain Heart broth (BH, Acumedia, Lansing, MI, USA), grown at 37 ± 1°C with shaking at

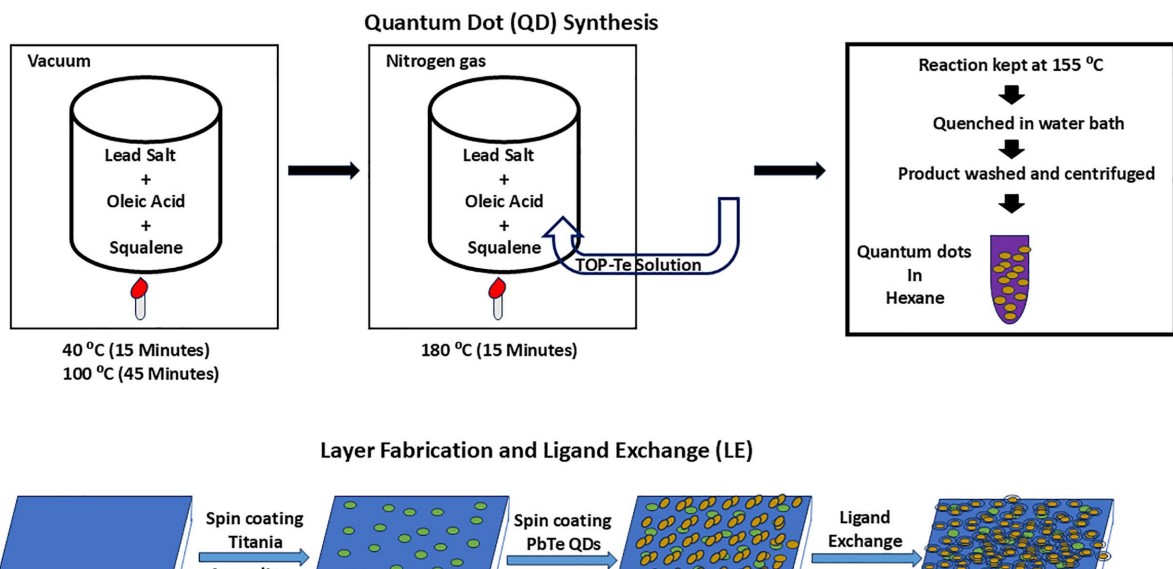

**Fig 1. A schematic diagram illustrating PbTe QDs synthesis and layer fabrication.**

170 rpm until reaching the absorbance of $OD_{660} \approx 0.3$, diluted with commercially available sterile 0.9% saline solution to $OD_{660} = 0.10 \pm 0.02$, which corresponded to a cell concentration of $10^8$ CFU mL$^{-1}$. The cells were diluted using saline to the final concentration of $10^6$ CFU mL$^{-1}$. Then, 0.1 mL of the bacterial suspension was spread on BH agar plates. Then the PbTe QD layers under investigation and blank controls (TiO$_2$/ITO glasses) were placed on plates, and the plates were incubated overnight at $37 \pm 1°C$ in the dark. For each test, the antibacterial activity was assessed by measuring the diameter of the zone of bacterial growth inhibition observed after incubation.

## 2.6. FTIR microscope measurement

FTIR microscope imaging was performed on an infrared microscope (Jasco, IRT-5200–16, Tokyo, Japan) coupled to an FTIR spectrometer (Jasco, 6800 FV, Tokyo, Japan). The 16x Cassegrain objective was used for all measurements in reflection mode and the lattice measurement option. The aperture was set to 50 μm x 50 μm per measurement point over a 6 x 6 square lattice. The radiation from the IR source of the spectrometer was focused on randomly selected points on the substrate, and the output radiation was reflected onto a liquid nitrogen cooled, mid-band MCT (mercury cadmium telluride) detector. Two measurements were carried out on each substrate in the spectral range of 650–4000 cm$^{-1}$ and each spectrum was an average over 32 scans. Before each measurement, a background was measured on a portion of the substrate observed to be clear of bacteria.

## 2.7. Data processing

All data processing procedures were carried out using the OriginPro software package (OriginPro Version 2023b, Origin-Lab Corporation, Northampton, MA, USA) and ImageJ software version 1.54f (Image Processing and Analysis in Java – Wayne Rasband and contributors, National Institutes of Health, Bethesda, MD, USA, public domain license, https://imagej.nih.gov/ij/). Spectra obtained from FTIR micro-spectroscopy were cut to the region 1000–1800 cm$^{-1}$, smoothed using the 18-side point Savitzky-Golay method, baseline corrected, and normalized to the peak around 1640 cm$^{-1}$ (C = O stretching vibration of amide I proteins) before further analysis.

## 3. Results

### 3.1. Characterization of PbTe layered substrates

The EDS spectrum of the PbTe nanoparticles presented in Fig 2A shows the presence of Pb (52.6%) and Te (47.4%) in the approximate stoichiometric ratio of PbTe compounds. The SEM image in Fig 2B displays 13.2 nm PbTe nanoparticles that are evenly distributed and have a cubic morphology. The smaller particles of 6.1 nm and 9.8 nm however, showed spherical morphology. Detailed characterization results for the proposed layers have been reported by Lyssenko et al. [33]. FTIR spectroscopy (Fig 3) shows transmittance peaks around wavenumbers that have been reported for PbTe nanocrystals synthesized through various routes. The peaks at 2852 cm$^{-1}$ and 2921 cm$^{-1}$ have been observed in electro-deposited PbTe films and attributed to anti-stretching –CH$_2$ and stretching –CH vibration modes, respectively [36]. The peaks around 1377 cm$^{-1}$ and 1462 cm$^{-1}$ were assigned to in-plane deformations in –CH$_3$ and –CH$_2$ respectively, while that observed at 2955 cm$^{-1}$ is due to asymmetric –CH$_3$ stretch [37]. The absorptions between 800 cm$^{-1}$ and around 1000 cm$^{-1}$ were attributed to –CH$_2$ rocking modes [36]. Fig 4 shows results of UV-Vis transmission measurements on PbTe layer compared to TiO$_2$/ITO substrate, indicating that for up to five layers of PbTe quantum dots, the coatings are largely transparent especially between 400 nm to 1100 nm, which is considered an important optical absorption region, for example, in silicon based solar cells.

The contact angle of double-distilled water on the surfaces before and after layering showed values that were significantly higher for the layered substrates than the blank controls (p < 0.01). The mean contact angles on the blank controls, the PbTe layers with LE and the PbTe layers without LE were found to be $61.5 \pm 2.0$, $82.8 \pm 4.8$, and $82.1 \pm 5.2$ degrees, respectively. S2 Fig shows examples of the contact angle measurement for the different surfaces.

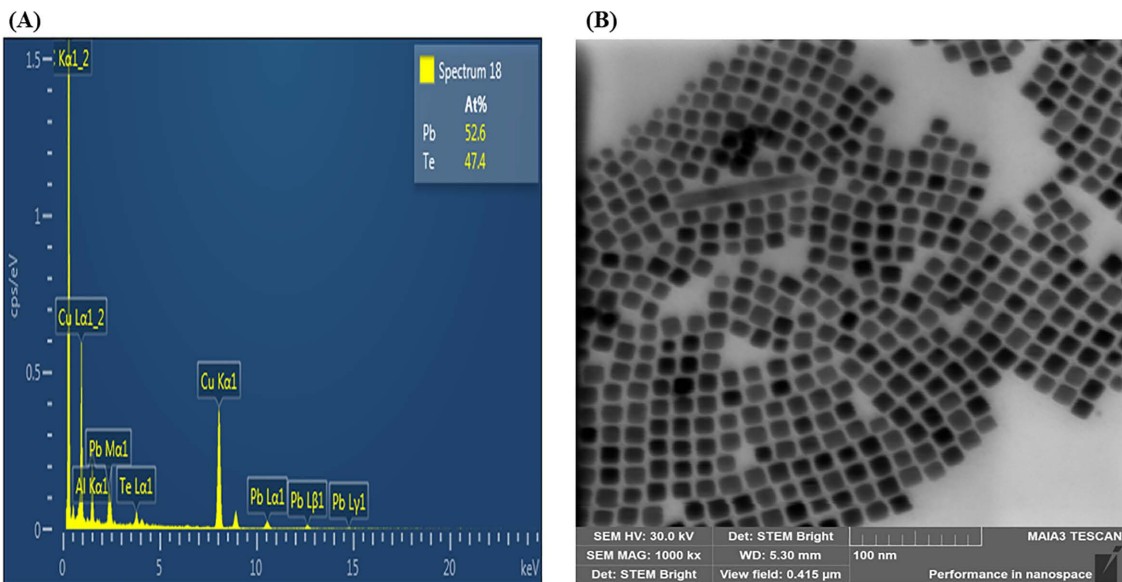

**Fig 2. EDS and SEM image of PbTe QD layer on TiO₂/ITO Glass. (A)** EDS spectrum, **(B)** SEM image.

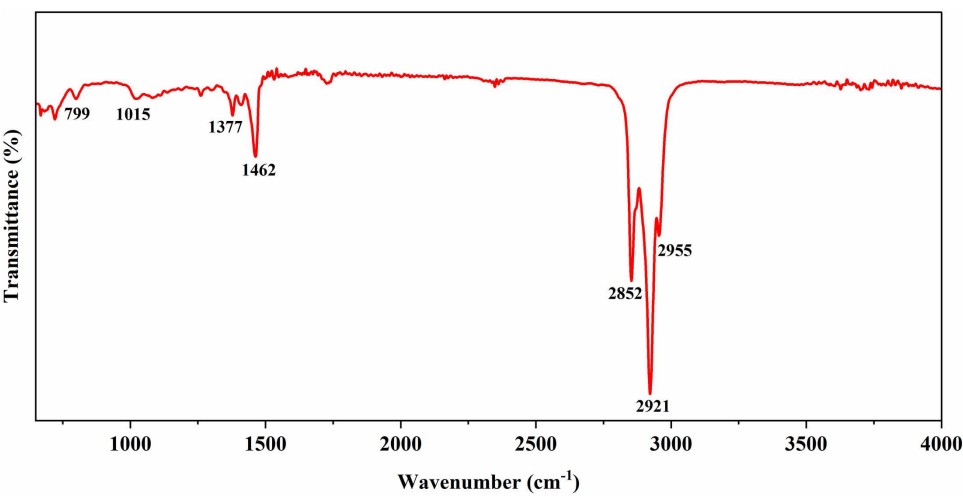

**Fig 3. FTIR spectrum of PbTe QDs.**

## 3.2. Zone of inhibition of PbTe QDs layered ITO glass substrates

The antibacterial efficacy of PbTe QD layers, fabricated on TiO₂/ITO glass substrates, was evaluated against Gram-positive *Staphylococcus aureus* and Gram-negative *Escherichia coli*, *Salmonella Paratyphi B* and *Pseudomonas aeruginosa*. The zone of inhibition (ZOI) for each bacteria type was determined using the PbTe layers, with and without LE. Blank TiO₂/ITO glasses of the same size were used as control. Fig 5 shows an example of the observed ZOI for 6.1 nm PbTe QD layers tested on the four bacterium types used in this study. It can be seen this material provides well-defined growth inhibition zones, except for *S. aureus*. The control did not show the inhibition effect on bacteria. Also, the ITO glass only did not show the antibacterial effect (S3 Fig). A summary of the diameters of the ZOI is presented in Fig 6. The

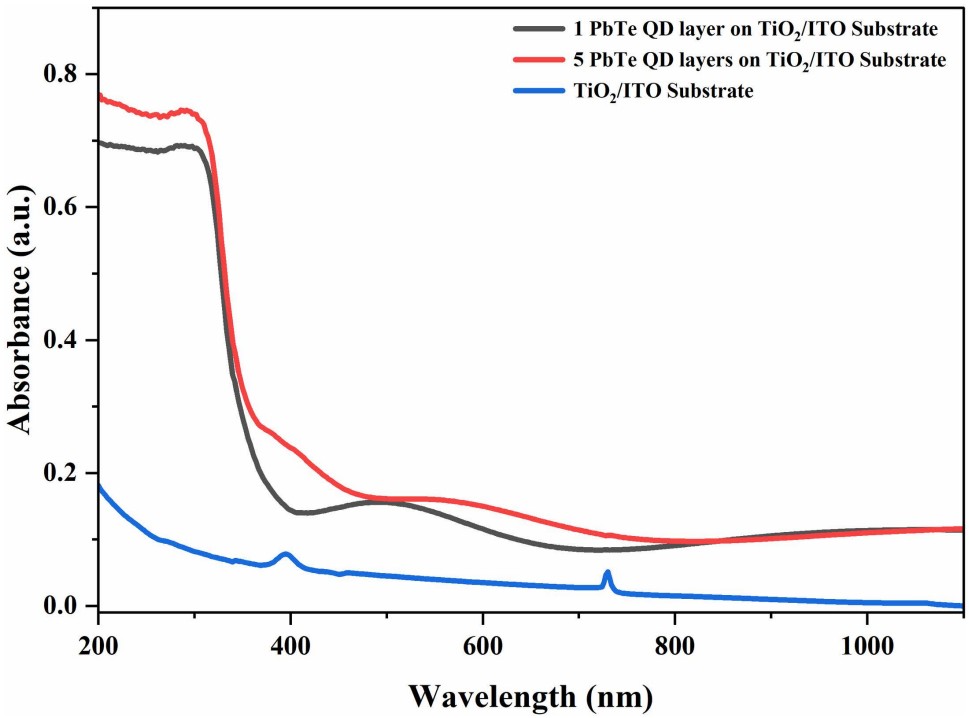

**Fig 4. UV-Vis transmission spectra for different number of layers of PbTe QDs on TiO$_2$/ITO glass.**

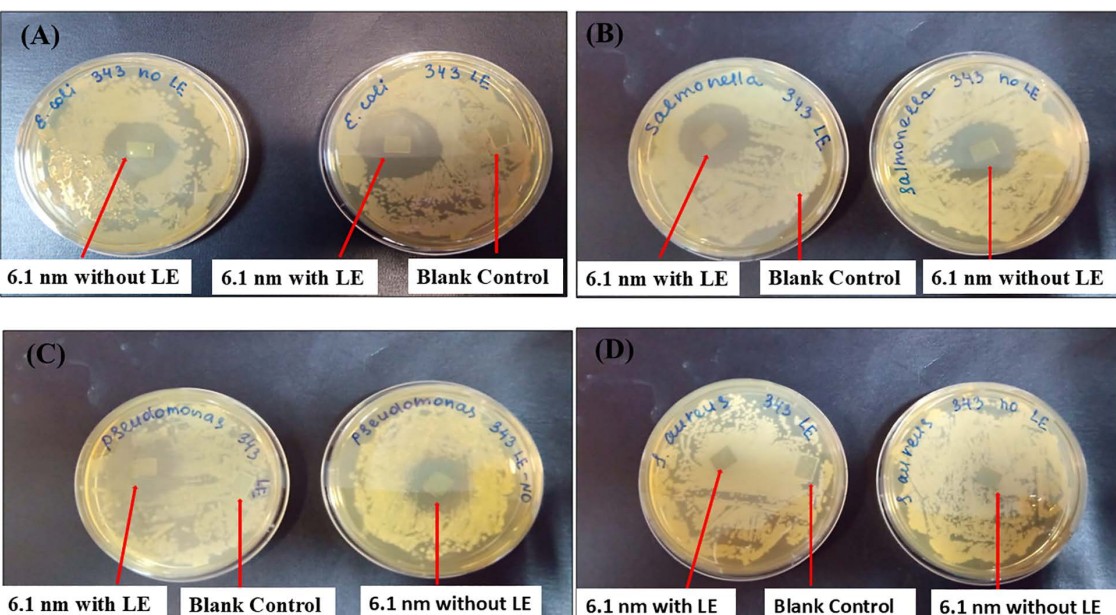

**Fig 5. Experiments showing antibacterial effect of 6.1 nm PbTe QD layers and blank controls. (A)** *E. coli,* **(B)** *S. Paratyphi B,* **(C)** *P. aeruginosa,* and **(D)** *S. aureus.*

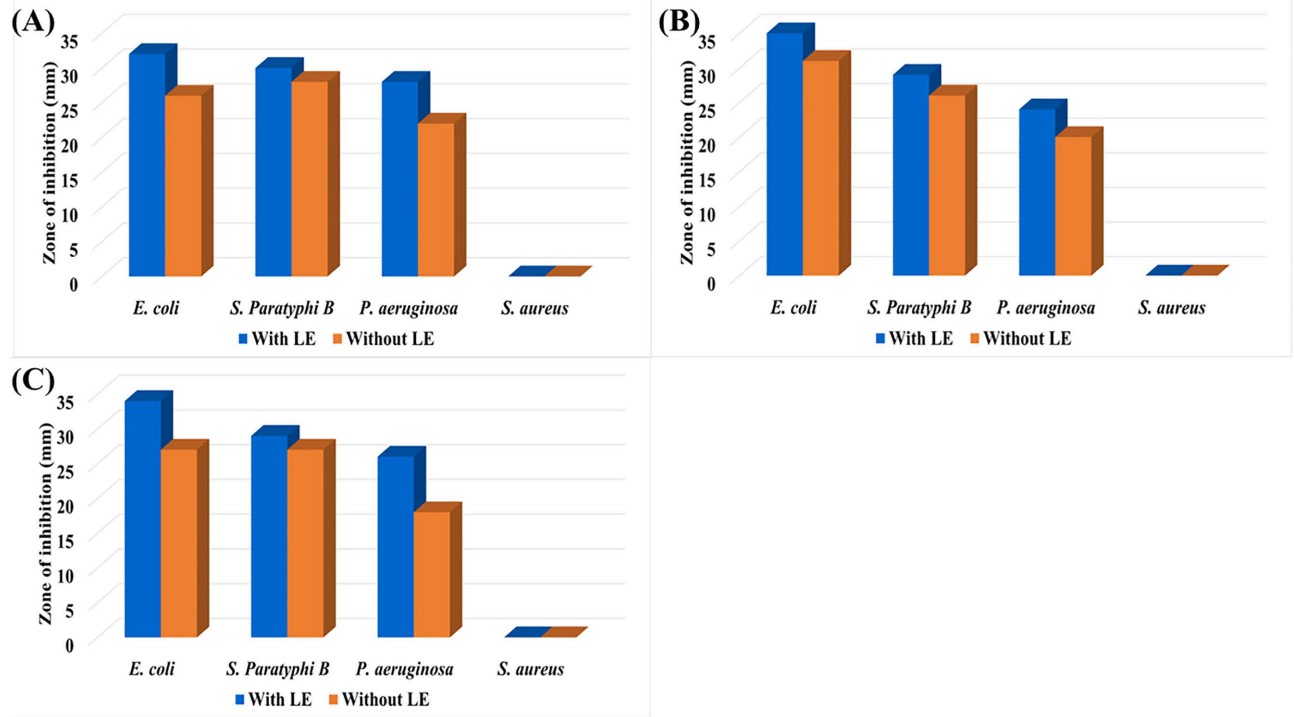

**Fig 6. Zone of inhibition of different sizes of PbTe QD layer. (A)** 6.1±0.5 nm, **(B)** 9.8±0.7 nm, and **(C)** 13.2±1.1 nm. *Measurement errors associated with the zones of inhibition are ±1 mm.*

diameter values ranged from 26–35 mm for *E. coli*, 26–30 mm for *S. Paratyphi B*, and 18–26 mm for *P. aeruginosa*. The effect was between negligible and sparse for the *S. aureus*. Layer functionalization with the organic ligand was observed to increase slightly the antibacterial properties of the PbTe QDs.

### 3.3. FTIR analysis of bacteria on PbTe QDs layers

To compare the mid-IR absorption profiles of the bacteria on the control and layered substrates, measured spectra were normalized to the amide I peak around 1640 cm$^{-1}$. Examples of the normalized spectra for each bacteria type in the region 1000−1800 cm$^{-1}$ are presented in Fig 7. On Fig 7A-C, it can be observed that for the bacteria on the PbTe QDs layers, there is a significant increase in the peaks around 1080 cm$^{-1}$ and 1400 cm$^{-1}$ compared to the blanks, and the peak around 1150 cm$^{-1}$ which is almost absent in the spectra of bacteria on the blank substrates. These peaks are, respectively, related to symmetric $PO_2^-$ vibrations in nucleic acids, symmetric $COO^-$ vibrations in amino acids, and phosphodiester vibrations [38]. They relate to RNA and DNA contents, and the observed increase could suggest a disruption of cell membrane integrity leading to leakage of cytoplasmic contents, including nuclear materials, and ultimately an increased infrared absorption on these bands.

## 4. Discussion

Semiconductor quantum dots including CdTe [39,40], CdSe [18], and PbS [41] have previously been investigated for their antimicrobial properties with results showing significant effect on a variety of organisms including bacteria and fungi. However, their applications, especially in biomedicine, are limited due to concerns for heavy metal toxicity as well as environmental friendliness [8]. Nevertheless, encapsulation and surface functionalization have been demonstrated to

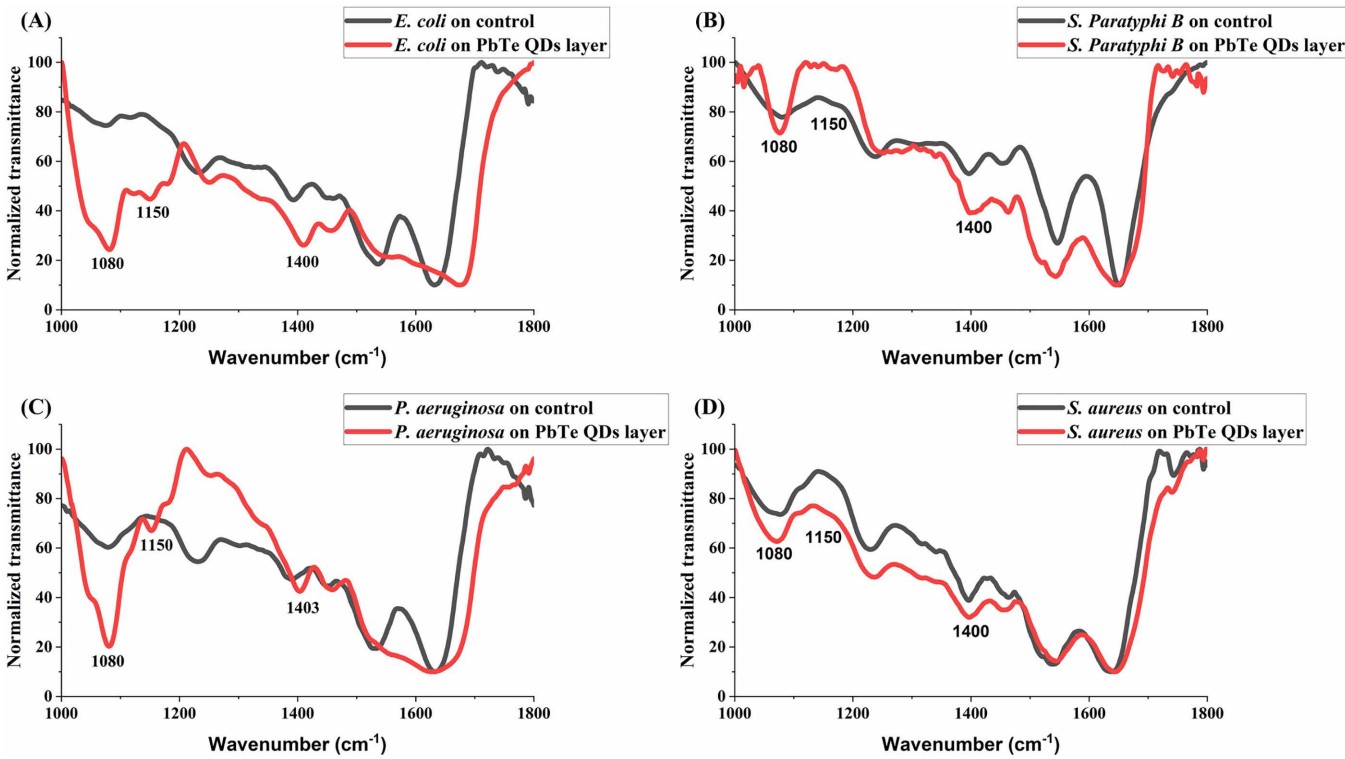

**Fig 7. Comparison of sample FTIR transmittance spectra of bacteria on 6.1 nm PbTe QD-layered substrates to those on blank controls. (A)** *E. coli,* **(B)** *S. Paratyphi B,* **(C)** *P. aeruginosa,* and **(D)** *S. aureus.*

significantly mitigate these issues, thereby enhancing the potential of nanomaterials for industrial and biomedical uses. Moreso, tailored applications such as demonstrated in the use of copper/nickel nano films on titanium condenser tubes of breeder reactors [42] provide a context wherein the advantage of antibacterial nanomaterials could be harnessed with little concern for biotoxicity. Similarly, ligand-functionalized PbTe QDs here investigated showed both antibacterial efficacy as well as significant transparency to solar spectra within the visible region and could be considered for anti-fouling coating on solar panels where they can reduce both cleaning requirements and maintenance cost. While our investigation is primarily an experimental proof of concept, for practical purposes, cost -to -benefit considerations for the product in terms of the proposed method, materials, and long-term durability of the coating are important and must be weighed accordingly for any specific applications.

The tested PbTe QDs layers showed notable antibacterial effect on Gram-negative *Escherichia coli*, *Salmonella Paratyphi B*, and *Pseudomonas aeruginosa*. However, a negligible effect was observed for Gram-positive *Staphylococcus aureus*. Researchers have proposed various mechanisms of antibacterial activity in the literature, such as cell wall interaction, membrane penetration, reactive oxygen species (ROS) production, DNA damage, and protein synthesis inhibition [11], which lead to the disruption of molecular processes and ultimately result in cell death. Here, we hypothesize the release of $Pb^{2+}$ ions and their diffusion, followed by electrostatic interaction with the bacterial cell wall, typically carrying a negative charge [12], attacking it and causing disruption of cytoplasmic boundaries, cell content leakage, and bacteria death. This hypothesis follows from the intense increase in normalized FTIR absorption intensities we observed on wavenumbers related to absorptions of nuclear material for bacteria found on the PbTe QD layers. Disruption of the cellular boundaries would mean the release of materials of the cytoplasm, including the nucleoid, making absorptions due to nuclear materials (RNA and DNA) more visible to the detector.

Damage to bacterial membrane has been estimated by the amount of DNA and RNA released from inner cellular constituents of bacteria, monitored using the UV absorbance peak at 260 nm [43]. The increased FTIR peaks in our study, especially the absorption around 1080 cm$^{-1}$ may serve a similar purpose, supporting the proposed mechanism of antibacterial action. Another mechanism that may result in the observed increase is surface plasmon resonant (SPR) absorption due to the quantum dots. However, if the increase was due to SPR alone, it should be observed also for the Gram positive bacteria (Fig 7D) on the PbTe layered substrate. Gram-positive bacteria cell walls are known to be thicker (about 20–80 nm) than those of Gram-negative (about 5–10 nm). They also have highly negatively charged polymers known as teichoic acid that aid cation sequestering and are covalently linked to the thick peptidoglycan layer of the cell walls. Gram negative bacteria on the other hand lack teichoic acid but have characteristic outer lipopolysaccharide membrane [44,45]. These structural and compositional differences could explain the difference in response of the bacterial cell types to stress such as that from PbTe quantum dots in our experiments. An additional mechanism which might be considered plausible is hydrogen peroxide induced oxidative stress as proposed in [46] for Pb chalcogenides; and in [41] for PbS nanopowders, where the intrinsic defects generate electron-hole pairs that interact with water and dissolved oxygen in a series of reactions to produce hydrogen peroxide. ROS generation reinforces and could help maintain the observed antibacterial activity in a setting where leaching of ionic lead is traded for more permanent immobilization of the active layer. The contact angles of water on layered substrates were found to be significantly greater than those on blank substrates (p < 0.01), indicative of a decrease in hydrophilicity/wettability of the substrates upon layering with the PbTe QDs. A decrease in wettability implies less likelihood of adhesion of soiling materials. Nevertheless, after each experiment, we observed attachment of dead bacteria to the layer surface. This can reduce the effectiveness of the active surface of PbTe layers by providing a new favorable layer for bacterial growth. However, we found that by autoclaving at 120°C, the antibacterial effect of the layers is repeated. S1 Fig shows the antibacterial effect observed after autoclaving layered substrates.

Ligand functionalization was found to improve the chemical stability of the QDs and the antibacterial properties of the layers. Due to the presence of lead in our QDs, and the fact that there are no safe levels of lead exposure [47], PbTe quantum dots are considered toxic, and may be used only for non-biomedical purposes. Another limitation is that of mechanical stability, especially to weathering or cleaning, which still needs to be addressed, and this could be a subject of further research for this material.

## Supporting information

**S1 Fig.  6.1 nm PbTe quantum dot layered substrates showing repeat of antibacterial effect (A) First contact, (B) post autoclave.**
(PDF)

**S2 Fig.  An example of the contact angle measurement on (A) PbTe quantum dot layer with LE, (B) PbTe quantum dot layer without LE, and (C) blank control.**
(PDF)

**S3 Fig.  An experiment showing the effect of ITO glass on representatives of Gram negative and Gram positive bacteria (A) *E. coli,* (B) *S. aureus*.**
(PDF)

## Author contributions

**Conceptualization:** Ayan Barbora, Refael Minnes.

**Data curation:** Samuel Onuh Abuh, Svetlana Lyssenko, Iryna Hovor.

**Formal analysis:** Samuel Onuh Abuh, Svetlana Lyssenko.

**Funding acquisition:** Faina Nakonechny, Refael Minnes.

**Investigation:** Samuel Onuh Abuh, Svetlana Lyssenko, Iryna Hovor.

**Methodology:** Faina Nakonechny, Refael Minnes.

**Project administration:** Ayan Barbora, Faina Nakonechny, Refael Minnes.

**Resources:** Faina Nakonechny, Refael Minnes.

**Supervision:** Faina Nakonechny, Refael Minnes.

**Validation:** Faina Nakonechny, Refael Minnes.

**Visualization:** Samuel Onuh Abuh, Svetlana Lyssenko.

**Writing – original draft:** Samuel Onuh Abuh.

**Writing – review & editing:** Samuel Onuh Abuh, Svetlana Lyssenko, Ayan Barbora, Iryna Hovor, Faina Nakonechny, Refael Minnes.

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
