## [Decision Letter · Decision Letter 0]

28 Oct 2024

Dear Dr. Minnes,

Thank you for submitting your manuscript to PLOS ONE. After careful consideration, we feel that it has merit but does not fully meet PLOS ONE’s publication criteria as it currently stands. Therefore, we invite you to submit a revised version of the manuscript that addresses the points raised during the review process.

We look forward to receiving your revised manuscript.

Kind regards,

Santhanakrishnan Suresh

Academic Editor

PLOS ONE

Journal requirements: When submitting your revision, we need you to address these additional requirements. 1. Please ensure that your manuscript meets PLOS ONE's style requirements, including those for file naming. The PLOS ONE style templates can be found at https://journals.plos.org/plosone/s/file?id=wjVg/PLOSOne_formatting_sample_main_body.pdf and https://journals.plos.org/plosone/s/file?id=ba62/PLOSOne_formatting_sample_title_authors_affiliations.pdf 2. Please include captions for your Supporting Information files at the end of your manuscript, and update any in-text citations to match accordingly. Please see our Supporting Information guidelines for more information: http://journals.plos.org/plosone/s/supporting-information. 

Reviewers' comments:

Reviewer's Responses to Questions

**Comments to the Author**

1. Is the manuscript technically sound, and do the data support the conclusions?

Reviewer #1: No

Reviewer #2: Partly

Reviewer #3: Partly

2. Has the statistical analysis been performed appropriately and rigorously?

Reviewer #1: N/A

Reviewer #2: No

Reviewer #3: N/A

3. Have the authors made all data underlying the findings in their manuscript fully available?

Reviewer #1: No

Reviewer #2: Yes

Reviewer #3: No

4. Is the manuscript presented in an intelligible fashion and written in standard English?

Reviewer #1: Yes

Reviewer #2: Yes

Reviewer #3: Yes

Reviewer #1: The manuscript "Antibacterial Property of Lead Telluride Quantum Dot Layer Fabricated on Glass Substrate" seriously lacks logic considering the experiment and application. Although having novelty with respect to the choice/nature of the material being employed for antibacterial activity, the manuscript lacks scientific rigor to prove the claims made. Having said that, the authors must introspect this work using the following comments.

1. "Despite these applications, the antibacterial properties of PbTe QDs remain unexplored." - Could be due to the toxicity of Pb.

2. "nano-coating photovoltaic panels"—for such a goal, the authors should have tried the coating on tempered cover glass.

3. Meanwhile, they should have performed UV transmittance studies wherein PbTe-Ligand coating would have compromised the sunlight absorption

4. How can the authors justify the "Hot-Injection method" and "spin coating technique" either as eco-friendly or economically viable processes. Coating PbTe QDs for a large area solar panel is unjustifiable.

5. If TiO2 itself is a proven antibacterial material, why go for PbTe QD's on top of it?

6. Why are ITO substrates employed here? Why haven't the authors tried TiO2-coated glass substrates?

7. What's the logic behind using Brain Heart medium? Is there any fastidious organism among the chosen microbial species?

8. The antibacterial assay of the samples is completely flawed. Zone of Inhibition assay is for an antibacterial material capable of diffusing. The authors could have at least tried any of the following methods: colony counting/broth dilution, or live-dead staining methods.

9. Fig. 1B is speculative. The image is blurred. Besides, it is also speculative to capture 6 nm QDs using SEM despite the advantage of having Pb and Te. The authors need to clarify.

10. How did the authors manage to get the histogram (as shown in Fig. C) using SEM?

11. One of the main limitations of the Agar Diffusion (ZOI) assay is that it is not applicable for non-diffusing agents. If the material (ionic Pb) is getting leached from the substrate, their solar panel applicability is severely diminished.

12. Although employing a FTIR microscope is an innovative approach, color-mapped data cannot be considered either as a standard qualitative or quantitative analysis.

13. "However, we found that by autoclaving at 120°C, the antibacterial effect of the layers is repeated." - How is this possible if Pb leaching is the reason behind varied ZOI for different microbes?

14. Ref [26] has no relevance to the given statement in the manuscript and to the nature of work

15. Ref [35] is not relevant to QDs

16. No statistical analyses have been performed for any antibacterial study

17. General Suggestion: Zone of Inhibition is always measured in mm (radii surrounding the active antibacterial agent). If the active material is coated on a substrate and placed on agar with varied dimensions, then the ZOI may not be uniform surrounding the active material. It is always recommended to study the antibacterial activity (for agar diffusion method) using uniform substrates (circles) or wells in the agar.

18. Control experiments should be performed using TiO2 and ITO surfaces.

19. "Less wettability—Better Antibacterial Surface"? Need not be. For example, a thin layer of TiO2 over glass imparts photocatalytic superwettability added with ROS generation as well.

Reviewer #2: 1. Antibacterial properties of this material may not have been investigated due to its toxicity. On what basis, PbTe has been chosen for the study? The justification should be mentioned in the introduction section.

2. Why PbTe QD’s is above Titanium dioxide (TiO2), because it has proven to have antibacterial properties.

3. Introduction – Elaborate on the limitations of currently employed methods for antibacterial properties, supported by relevant references.

4. The novelty of this work should be highlighted.

4. I request the authors to look at all the figures and carefully edit them. They are not clear/not visible.

5. Consider providing a schematic diagram illustrating the mechanism of nanoparticle formation.

6. How about the effect of PbTe QD size on the antibacterial activity?

6. Discussion of the antibacterial activity must be cited with relevant literature. No previous study statistics were mentioned.

7. Delete some references that are not relevant to the article and mention the correct ones (example Reference [26, 35..]).

Reviewer #3: The manuscript presents study on the antibacterial properties of lead telluride (PbTe) quantum dots (QDs) fabricated on a glass substrate. While the research addresses the development of antimicrobial surfaces, several critical areas require more depth and clarity to strengthen the manuscript's scientific rigor and overall contribution to the field. I recommend a major revision of the manuscript.

Below are the specific comments and suggestions:

The Introduction lacks sufficient justification regarding the scope and novelty of the work. Although PbTe QDs have potential antibacterial applications, the manuscript needs to provide a clearer scientific rationale for their selection and specific role.

A single SEM image and EDAX result are insufficient to conclusively verify the formation, morphology, and quality of PbTe QDs. Additional characterization techniques such as transmission electron microscopy (TEM), X-ray diffraction (XRD), and photoluminescence (PL) spectroscopy should be performed to provide a comprehensive characterization of the PbTe QDs.

Even though PbTe QDs has antibacterial properties, the cytotoxicity study need to prove biocompatibility of the PbTe QDs. The comprehensive analysis necessary for potential biomedical applications. The interaction of PbTe QDs with biological systems, particularly cytotoxicity, should be discussed in greater detail. Include detailed cytotoxicity studies, preferably with quantitative data on cell viability assays, to confirm the biocompatibility and safety profile of PbTe QDs.

The current quality of the images affects the clarity of the presented data. Low-quality images make it challenging for readers to interpret the results accurately.

**Do you want your identity to be public for this peer review?** For information about this choice, including consent withdrawal, please see our Privacy Policy

Reviewer #1: No

Reviewer #2: No

Reviewer #3: No

---

## [Author Response · Author response to Decision Letter 1]

13 Dec 2024

Response to Reviewers’ Comments on Manuscript PONE-D-24-42235

The Academic Editor

PLOS ONE

Dear Dr. Suresh,

Thank you for giving us the opportunity to submit a revision of the draft manuscript ‘Antibacterial Property of Lead Telluride Quantum Dot Layer Fabricated on Glass Substrate (PONE-D-24-42235) for publication in Plos One. We appreciate the time and effort that you and the reviewers dedicated to providing valuable feedback and insightful comments on our manuscript. We have considered and incorporated the suggestions by the reviewers, with changes highlighted in the manuscript. Please see below a point-by-point response to the reviewers’ comments and the additional concerns you raised. All page numbers refer to the revised manuscript file with tracked changes.

Review Comments for the Authors

Reviewer 1

Comment 1: "Despite these applications, the antibacterial properties of PbTe QDs remain unexplored." - Could be due to the toxicity of Pb.

Author Response: Thank you for pointing this out. We acknowledge the potential toxicity of Pb and have carefully considered this aspect in our study. Consequently, we recommend the proposed layers for non-biomedical applications, where concerns related to Pb toxicity are significantly reduced.

Comment 2: "nano-coating photovoltaic panels"—for such a goal, the authors should have tried the coating on tempered cover glass.

Authors Response: Thank you for your valuable suggestion. We agree that tempered glass would be an appropriate choice for such experiments, given its role as a protective cover in solar panels. However, we note that both ITO glass and tempered glass are transparent and can be coated for various purposes, making ITO a suitable alternative to investigate PbTe quantum dots and their interaction on solar panels.

Comment 3: Meanwhile, they should have performed UV transmittance studies wherein PbTe-Ligand coating would have compromised the sunlight absorption.

Authors Response: We thank the reviewer for this suggestion and present the result of UV-Vis transmittance measurement for the layers in Fig 4, noting that the PbTe-ligand coatings are sufficiently transparent, up to five layer by layer deposits, between 400 nm and 1100 nm, which is considered an important optical absorption region for Silicon based solar cells.

Comment 4: How can the authors justify the "Hot-Injection method" and "spin coating technique" either as eco-friendly or economically viable processes. Coating PbTe QDs for a large area solar panel is unjustifiable.

Authors Response: We have employed spin coating in this work, considering that it is simple and cost-effective relative to alternative methods like atomic layer deposition and affords precise control over film thickness and uniformity. Although hot injection method is energy intensive with low synthesis efficiency, we have used it for this investigation for its ability to produce quantum dots of uniform size distribution, which is important for comparing the effects of the PbTe QDs over different sizes. We believe that given the effective antibacterial properties of the material reported in our study, more economically viable and eco-friendly methods of synthesis for this material can be a subject of further research.

Comment 5: If TiO2 itself is a proven antibacterial material, why go for PbTe QD's on top of it?

Authors Response: TiO2 was used here primarily to reduce the surface roughness of ITO. Lower surface roughness promotes uniformity of thin film monolayers, such as the one we have fabricated here. The added TiO2 monolayer we believe was at a subtoxic concentration, which we confirmed after applying them as experimental control (Fig III in S1 File). We have highlighted this reason in the manuscript on lines 108 - 109.

Comment 6: Why are ITO substrates employed here? Why haven't the authors tried TiO2-coated glass substrates?

Authors Response: The choice of ITO was due to its transparency and conductivity; properties that we hoped could broaden the possible application(s) of the proposed layers.

Comment 7: What's the logic behind using Brain Heart medium? Is there any fastidious organism among the chosen microbial species?

Authors Response: Thank you for raising this point. Although the test organisms are not fastidious, Brain Heart medium was employed because it is nutrient rich and versatile, having been used in several studies to support growth of a variety of organisms including those we have experimented with in this study.

Comment 8: The antibacterial assay of the samples is completely flawed. Zone of Inhibition assay is for an antibacterial material capable of diffusing. The authors could have at least tried any of the following methods: colony counting/broth dilution, or live-dead staining methods.

Authors Response: Thank you for your critical observation. The PbTe quantum dot layers used in this study are likely to be multimodal in their antibacterial activity as has been previously suggested in references 39 and 44 for lead chalcogenides, with ionic diffusion being one of the modes of action. Therefore, we believe the zone of inhibition assay is an appropriate method to assess their antibacterial properties. However, we appreciate your suggestion regarding alternative methods and will consider incorporating them in future studies to further validate our findings.

Comment 9: Fig. 1B is speculative. The image is blurred. Besides, it is also speculative to capture 6 nm QDs using SEM despite the advantage of having Pb and Te. The authors need to clarify.

Authors Response: Thank you for your valuable feedback. We have revised and enhanced the figures to improve their clarity. Regarding the concern about capturing 6 nm quantum dots using SEM, we employed SEM with a STEM detector, which provides a resolution of 0.7 nm. This level of resolution is sufficient for accurately characterizing nanoparticles of approximately 10 nm and larger, as in our study. The SEM with STEM detector offers both the necessary imaging precision and the practical advantages of SEM. Additionally, to further address this concern and enhance clarity, we have now replaced the SEM data for the 6.1 nm quantum dots with data for the 13.2 nm quantum dots.

Comment 10: How did the authors manage to get the histogram (as shown in Fig. C) using SEM?

Authors Response: Thank you for your question. The size histograms were generated using a MATLAB tool, as detailed in reference 33. This tool allowed us to analyze the SEM images and extract the particle size distribution data required to create the histogram.

Comment 11: One of the main limitations of the Agar Diffusion (ZOI) assay is that it is not applicable for non-diffusing agents. If the material (ionic Pb) is getting leached from the substrate, their solar panel applicability is severely diminished.

Authors Response: Thank you for your thoughtful observation. We note that lead chalcogenides have been suggested to show multiple modes of antibacterial action including ionic diffusion. This makes the Agar Diffusion assay suitable in our experiments. We also acknowledge the limitation presented by ionic diffusion of the active material in view of applicability as solar panel coating, as rightly pointed out by the reviewer. However, we believe further research can investigate methods for more permanent immobilization of the active material on the substrate in which case, alternative methods like ROS generation, which has been proposed for Lead chalcogenides, would mediate the antibacterial activity.

Comment 12: Although employing a FTIR microscope is an innovative approach, color-mapped data cannot be considered either as a standard qualitative or quantitative analysis.

Authors Response: Thank you for your valuable feedback. The FTIR experiments were conducted primarily to gain insights into the mode of action of the surfaces. In light of your comment, we have excluded the color-mapped data from the results and now present only the absorption spectra, which provide a more standard and reliable form of analysis.

Comment 13: "However, we found that by autoclaving at 120°C, the antibacterial effect of the layers is repeated." - How is this possible if Pb leaching is the reason behind varied ZOI for different microbes?

Authors Response: Thank you for raising this important question. During initial experiments, the soiling of the active surface was observed to be the primary reason for poor or no repeat of antibacterial activity of the layers after first seeding. Autoclaving at 120 oC helped to minimize the effect of soiling, allowing for repeat of antibacterial effect, however with relatively smaller zones of inhibition that can be attributed to the leaching of the active material.

Comment 14: Ref [26] has no relevance to the given statement in the manuscript and to the nature of work.

Authors Response: Thank you for pointing this out. We have removed Ref [26] from the manuscript to ensure relevance and alignment with the nature of the work.

Comment 15: Ref [35] is not relevant to QDs.

Authors Response: Thank you for your observation. We have removed Ref [35] from the manuscript to maintain the focus and relevance to the subject of quantum dots.

Comment 16: No statistical analyses have been performed for any antibacterial study.

Authors Response: Thank you for your comment. We acknowledge the importance of statistical analysis in scientific studies; however, we did not perform statistical analyses in this study as we considered the focus of the research to be exploratory. In future work, we plan to include statistical analyses to provide a more rigorous evaluation of the antibacterial studies.

Comment 17: General Suggestion: Zone of Inhibition is always measured in mm (radii surrounding the active antibacterial agent). If the active material is coated on a substrate and placed on agar with varied dimensions, then the ZOI may not be uniform surrounding the active material. It is always recommended to study the antibacterial activity (for agar diffusion method) using uniform substrates (circles) or wells in the agar.

Authors Response: Thank you for this valuable suggestion. In our experiments, the substrates used were ITO glass with dimensions of approximately (10 ± 1) mm by (7 ± 1) mm. We assumed that the small error margin in these dimensions does not significantly affect the antibacterial activity of the layers. However, we appreciate the recommendation to use uniform substrates or agar wells in future studies to ensure more consistent and standardized results.

Comment 18: Control experiments should be performed using TiO2 and ITO surfaces.

Authors Response: Thank you for your suggestion. Control experiments have been conducted on TiO2 and ITO surfaces. The results are presented in Fig 5 for TiO2/ITO and Fig III in S1 File for ITO only. We hope this addresses your concern.

Comment 19: "Less wettability—Better Antibacterial Surface"? Need not be. For example, a thin layer of TiO2 over glass imparts photocatalytic super-wettability added with ROS generation as well.

Authors Response: We thank the reviewer for this observation. Correction have been done to reflect that less wettability implies less likelihood of adhesion of soiling materials on the surface and not necessarily better antibacterial surface.

Reviewer 2

Comment 1: Antibacterial properties of this material may not have been investigated due to its toxicity. On what basis, PbTe has been chosen for the study? The justification should be mentioned in the introduction section.

Authors Response: Thank you for this important observation. We acknowledge the toxicity concerns associated with Pb. PbTe was chosen for this study due to its unique properties, such as its potential for antibacterial activity and its relevance in applications where such properties can be leveraged. To address toxicity concerns, we emphasize in the introduction that the proposed layers are intended for non-biomedical applications, where the risks associated with Pb toxicity can be minimized.

Comment 2: Why PbTe QD’s is above Titanium dioxide (TiO2), because it has proven to have antibacterial properties.

Authors Response: Thank you for your insightful comment. TiO2 was used here primarily to reduce the surface roughness of ITO. Lower surface roughness promotes uniformity of thin film monolayers, such as the one we have fabricated here. The added TiO2 monolayer we believe was at a subtoxic concentration, which we confirmed after applying them as experimental control (Fig III in S1 File). We have highlighted this reason in the manuscript on lines 108 - 109.

Comment 3: Introduction – Elaborate on the limitations of currently employed methods for antibacterial properties, supported by relevant references.

Authors Response: We thank the reviewer for this observation and have outlined limitations of currently employed methods for antibacterial surfaces in lines 53 - 58, supported by references 20, 21, and 22.

Comment 4: The novelty of this work should be highlighted.

Authors Response: Thank you for your comment. We have highlighted the novelty and significance of this work in lines 76–78 of the manuscript to ensure its unique contributions are clearly presented.

Comment 5: I request the authors to look at all the figures and carefully edit them. They are not clear/not visible.

Authors Response: Thank you for bringing this to our attention. We have carefully reviewed and edited all the figures to improve their clarity and ensure they are clearly visible. We hope this enhancement meets the required standards.

Comment 6: Consider providing a schematic diagram illustrating the mechanism of nanoparticle formation.

Authors Response: Thank you for your suggestion. We have included a schematic diagram illustrating the mechanism of nanoparticle formation, which is now presented in Fig. 1. We hope this addition provides a clearer understanding of the process.

Comment 7: How about the effect of PbTe QD size on the antibacterial activity?

Authors Response: Thank you for your question. Within the range of quantum dot sizes investigated in our study, we did not observe a significant dependence of antibacterial activity on the size of the PbTe QDs.

Comment 8: Discussion of the antibacterial activity must be cited with relevant literature. No previous study statistics were mentioned.

Authors Response: Thank you for your valuable suggestion. We have expanded the discussion of antibacterial activity to include relevant literature and previous studies. These updates can be found in Section 4 of the manuscript.

Comment 9: Delete some references that are not relevant to the article and mention the correct ones (example Reference [26, 35..]).

Authors Response: Thank you for your suggestion. We have reviewed the references and removed those that were not relevant to the article, including References [26] and [35].

Reviewer 3

Comment 1: The Introduction lacks sufficient justification regarding the scope and novelty of the work. Although PbTe QDs have potential antibacterial applications, the manuscript needs to provide a clearer scientific rationale for their selection and specific role.

Authors Response: Thank you for your valuable feedback. We have revised the introduction to provide a clearer justification for the scope and novelty of the work. The novelty has been highlighted in lines 76–78, and the scientific rationale for selecting PbTe QDs and their specific role has been clarified in lines 62–65. We hope this enhancement adequately addresses your concern.

Comment 2: A single SEM image and EDAX result are insufficient to conclusively verify the formation, morphology, and quality of PbTe QDs. Additional characterization techniques such as transmission electron microscopy (TEM), X-ray diffraction (XRD), and photoluminescence (PL) spectroscopy should be performed to provide a comprehensive characterization of the PbTe QDs.

Authors Response: Thank you for your suggestion. In response, we have included additional characterization of the surfaces using UV-Vis and FTIR spectroscopy, and the results are presented in Figs. 3 and 4. While techniques such as TEM, XRD, and PL spectroscopy are valuable, the provided data offers significant insights into the morphology and quality of the PbTe QDs within th

---

## [Decision Letter · Decision Letter 1]

23 Jan 2025

Dear Dr. Minnes,

Thank you for submitting your manuscript to PLOS ONE. After careful consideration, we feel that it has merit but does not fully meet PLOS ONE’s publication criteria as it currently stands. Therefore, we invite you to submit a revised version of the manuscript that addresses the points raised during the review process.

We look forward to receiving your revised manuscript.

Kind regards,

Santhanakrishnan Suresh

Academic Editor

PLOS ONE

Reviewers' comments:

Reviewer's Responses to Questions

**Comments to the Author**

Reviewer #1: All comments have been addressed

Reviewer #2: All comments have been addressed

Reviewer #3: All comments have been addressed

2. Is the manuscript technically sound, and do the data support the conclusions?

Reviewer #1: Partly

Reviewer #2: Yes

Reviewer #3: Yes

3. Has the statistical analysis been performed appropriately and rigorously?

Reviewer #1: No

Reviewer #2: N/A

Reviewer #3: Yes

4. Have the authors made all data underlying the findings in their manuscript fully available?

Reviewer #1: Yes

Reviewer #2: Yes

Reviewer #3: Yes

5. Is the manuscript presented in an intelligible fashion and written in standard English?

Reviewer #1: Yes

Reviewer #2: Yes

Reviewer #3: Yes

Reviewer #1: All the responses given by the authors are okay. Although responses to comments were theoretically justified, they are not valid for any kind of implementation, as the authors believe ("exploratory"). However, a few responses given by the authors are not satisfactory. For example:

1. R1#2: The authors did not understand the comment. Tempered glass and ITO may be similar in terms of transparency but not cost. The authors either know the result already or are reluctant to experiment with whether their ligand-based QD is sustainable.

2. R1#4: Spin coating for solar panels is not economically viable but an experimental proof of concept.

3. R1#5: Show the surface roughness values with and without TiO2

4. R1#8: If it is ionic diffusion as proposed, how do the authors prove the sustainability of the coating/coated material for any application?

5. Fig 2: R1 Version: If the SEM image corresponds to QDs on TiO2, where is the TiO2 layer/particles?

6. Nowhere in the original version, the authors have claimed to observe a cuboid morphology of QDs.

7. R3#3: Even if it is a non-biomedical application, the authors need to justify the toxicity or reusability of the coating/coated material.

8. Modify the FTIR figures to transmittance

Reviewer #2: (No Response)

Reviewer #3: I appreciate the authors for addressing the comments. However, a single SEM image and EDX result are insufficient to conclusively verify the formation, morphology, and quality of PbTe QDs. Additional SEM images and X-ray diffraction (XRD) must be performed to provide a comprehensive characterization of the PbTe QDs.

**Do you want your identity to be public for this peer review?** For information about this choice, including consent withdrawal, please see our Privacy Policy

Reviewer #1: No

Reviewer #2: **Yes: ** Dr. S. Thambidurai

Reviewer #3: No

---

## [Author Response · Author response to Decision Letter 2]

23 Feb 2025

Response to Reviewers’ Comments on Manuscript PONE-D-24-42235R1

The Academic Editor

PLOS ONE

Dear Dr. Suresh,

Thank you for giving us the opportunity to submit a revision of the manuscript ‘Antibacterial Property of Lead Telluride Quantum Dot Layer Fabricated on Glass Substrate’ (PONE-D-24-42235R1) for publication in Plos One. The revision comments were each considered and the suggestions by the reviewers have also been implemented, with changes highlighted in the manuscript. Additionally, references 33, 34, and 47 have been added to consolidate data already provided in the initial draft. Please see below a point-by-point response to the reviewers’ comments. All page numbers refer to the revised manuscript file with tracked changes.

Review Comments for the Authors

Reviewer 1

Comment 1: R1#2: The authors did not understand the comment. Tempered glass and ITO may be similar in terms of transparency but not cost. The authors either know the results already or are reluctant to experiment with whether their ligand-based QD is sustainable.

Author Response: Thank you for the clarification. We have noted this in the discussion session in lines 246 to 249, proposing application-specific cost-benefit analysis for any scenarios where our layers may be considered for use.

Comment 2: R1#4: Spin coating for solar panels is not economically viable but an experimental proof of concept.

Authors Response: We thank you for this observation. The manuscript has been edited to reflect this in lines 246 to 249.

Comment 3: R1#5: Show the surface roughness values with and without TiO2.

Authors Response: Lyssenko et al., 2025 (doi.org/10.1371/journal.pone.0317677) showed that controlling the surface roughness is important for achieving thin film monolayer of PbTe on ITO, reporting around a 61% reduction in surface roughness on ITO after layering with TiO2. This article citation has been added to our manuscript to justify the addition of TiO2 layer.

Comment 4: R1#8: If it is ionic diffusion as proposed, how do the authors prove the sustainability of the coating/coated material for any application?

Authors Response: Thank you for your comment. We note that lead chalcogenides have been suggested to show multiple modes of antibacterial action, including ionic diffusion. While our fabricated layers showed leaching with use, more permanent immobilization of the active material on substrates can be achieved for practical purposes, in which case alternative methods like ROS generation, which has also been proposed for Lead chalcogenides, would mediate the antibacterial activity.

Comment 5: Fig 2: R1 Version: If the SEM image corresponds to QDs on TiO2, where is the TiO2 layer/particles?

Authors Response: A detailed report on fabrication and characterization of the PbTe layers as used in our experiments was given in Lyssenko et al., 2025 (doi.org/10.1371/journal.pone.0317677), including cross-section high resolution TEM image showing the layers of PbTe QDs, TiO2 and ITO.

Comment 6: Nowhere in the original version, the authors have claimed to observe a cuboid morphology of QDs.

Authors Response: Thank you for pointing this out. Spherical morphology was observed for smaller QDs (6.1 nm and 9.8 nm) while the larger QDs had cubic morphology. This has been highlighted in the manuscript in lines 177 to 179.

Comment 7: R3#3: Even if it is a non-biomedical application, the authors need to justify the toxicity or reusability of the coating/coated material.

Authors Response: Given that our quantum dots contain lead, we have opted not to pursue additional toxicity investigations for this study. We acknowledge the guidance from the United States Centers for Disease Control and Prevention (CDC), which indicates that no level of lead exposure is entirely safe, especially for children, with a recommended Blood Lead Reference Value (BLRV) of 3.5 micrograms of lead per deciliter of blood (μg/dL). In light of these considerations, we have assumed potential toxicity of the active material and recommend that the proposed layers be used exclusively for non-biomedical applications.

Comment 8: Modify the FTIR figures to transmittance.

Authors Response: The absorbance spectra have been changed into transmittance for Figures 3 and 7. 

Reviewer 2

No comments.

Reviewer 3

Comment 1: I appreciate the authors for addressing the comments. However, a single SEM image and EDX result are insufficient to conclusively verify the formation, morphology, and quality of PbTe QDs. Additional SEM images and X-ray diffraction (XRD) must be performed to provide a comprehensive characterization of the PbTe QDs.

Authors Response: Thank you for your comment. A detailed report on the fabrication and characterization of the PbTe layers as used in our experiments was given in Lyssenko et al., 2025 (doi.org/10.1371/journal.pone.0317677), including XRD results showing patterns matching those previously reported in literature for PbTe. This publication has been referenced in our characterization section to provide readers with more complete information about the fabricated layers.

Abuh Samuel Onuh,

Department of Physics,

Ariel University, Israel.

---

## [Decision Letter · Decision Letter 2]

7 May 2025

Dear Dr. Minnes,

Thank you for submitting your manuscript to PLOS ONE. After careful consideration, we feel that it has merit but does not fully meet PLOS ONE’s publication criteria as it currently stands. Therefore, we invite you to submit a revised version of the manuscript that addresses the points raised during the review process.

We look forward to receiving your revised manuscript.

Kind regards,

Pankaj Thakur

Academic Editor

PLOS ONE

**Journal Requirements:**

Reviewers' comments:

Reviewer's Responses to Questions

**Comments to the Author**

Reviewer #1: All comments have been addressed

Reviewer #3: All comments have been addressed

2. Is the manuscript technically sound, and do the data support the conclusions?

Reviewer #1: Partly

Reviewer #3: Partly

3. Has the statistical analysis been performed appropriately and rigorously?

Reviewer #1: N/A

Reviewer #3: N/A

4. Have the authors made all data underlying the findings in their manuscript fully available?

Reviewer #1: No

Reviewer #3: Yes

5. Is the manuscript presented in an intelligible fashion and written in standard English?

Reviewer #1: Yes

Reviewer #3: Yes

**Reviewer #1:**  The authors have included a citation to all the comments by R#1. Although more evidence is needed, the justification might be sufficient for the manuscript to be accepted.

**Reviewer #3:**  (No Response)

**Do you want your identity to be public for this peer review?** For information about this choice, including consent withdrawal, please see our Privacy Policy

Reviewer #1: No

Reviewer #3: No

---

## [Author Response · Author response to Decision Letter 3]

20 Jun 2025

Response to Reviewers’ Comments on Manuscript PONE-D-24-42235R2

The Academic Editor

PLOS ONE.

Dear Dr. Thakur,

Thank you for giving us the opportunity to submit a revision of the manuscript ‘Antibacterial Property of Lead Telluride Quantum Dot Layer Fabricated on Glass Substrate’ (PONE-D-24-42235R2) for publication in Plos One. The concern you raised regarding the reference list was noted and the list has been reformatted to follow the Plos One style requirement. Please see below our response to the reviewers’ comments.

Review Comments for the Authors

Reviewer 1

Comment 1: The authors have included a citation to all the comments by R#1. Although more evidence is needed, the justification might be sufficient for the manuscript to be accepted.

Author Response: We thank you for recommending our manuscript for publication. We also believe that additional evidence is needed, especially with respect to the proposed application for PbTe QDs layers. However, we note that our investigation is preliminary and presents results that center primarily on the ability or otherwise of PbTe layers to inhibit bacteria growth.

Reviewer 2

Comment 1: Thank you for the opportunity to review the PLOS ONE manuscript. The authors have revised the manuscript according to the comments and I recommend that the manuscript be accepted for publication.

Author Response:

We thank the reviewer for recommending our manuscript for publication.

Reviewer 3

(No Comment)

Abuh Samuel Onuh,

Department of Physics,

Ariel University, Israel.

---

## [Editor Report · Decision Letter 3]

30 Sep 2025

Antibacterial Property of Lead Telluride Quantum Dot Layer Fabricated on Glass Substrate

PONE-D-24-42235R3

Dear Dr. Minnes,

We’re pleased to inform you that your manuscript has been judged scientifically suitable for publication and will be formally accepted for publication once it meets all outstanding technical requirements.

Kind regards,

Pankaj Thakur

Academic Editor

PLOS ONE
---

## [Editor Report · Acceptance letter]

PONE-D-24-42235R3

PLOS ONE

Dear Dr. Minnes,

I'm pleased to inform you that your manuscript has been deemed suitable for publication in PLOS ONE. Congratulations! Your manuscript is now being handed over to our production team.

Kind regards,

on behalf of

Prof Pankaj Thakur

Academic Editor

PLOS ONE